# A Process for the Separation of Noble Metals from HCl Liquor Containing Gold(III), Palladium(II), Platinum(IV), Rhodium(III), and Iridium(IV) by Solvent Extraction

**Wei Dong Xing and Man Seung Lee \***

Department of Advanced Material Science & Engineering, Institute of Rare Metal, Mokpo National University, Chonnam 534-729, Korea; weizi314159@126.com

\* Correspondence: mslee@mokpo.ac.kr; Tel.: +82-61-450-2492

**Abstract:** The demand for noble metals is increasing, owing to their excellent chemical and physical properties. In order to meet the demand, the recovery of noble metals with high purity from diverse secondary resources, which contain small amounts of noble metals, is of immense value. In this work, the possibility of the separation of Au(III), Pd(II), Pt(IV), Rh(III), and Ir(IV) by solvent extraction from a synthetic HCl solution is investigated. Only Au(III) was selectively extracted by Cyanex 272 in the HCl concentration range from 0.5 M to 9 M, leaving the other metal ions in the raffinate. The loaded Au(III) in Cyanex 272 was efficiently stripped by $(NH_2)_2CS$. The other four noble metals were sequentially separated on the basis of the procedures reported in the previous work. The mass balance showed that about 98% of each metal, except Pt(IV), was recovered by the proposed process. An efficient process for the recovery of the five noble metal ions from the HCl leaching solution of secondary resources containing these metals can be developed.

**Keywords:** solvent extraction; gold; platinum group metals; HCl

## 1. Introduction

Noble metals are indispensable for the manufacture of advanced materials, which are employed in the automobile, chemical, electronic, and medical industries, owing to their excellent chemical and physical properties [1,2]. Considering the demand for these metals and environmental protection, the secondary resources resulting from the above-mentioned industries have attracted much attention [3–5].

Many processes have been proposed for the recovery of noble metals from secondary resources [1]. Since the contents of noble metals in the secondary resources are very low, smelting of these secondary resources together with other metal ores is effective in recovering noble metals. For example, diverse secondary resources containing noble metals can be treated by a smelting operation in a copper smelter when the secondary resources do not contain components harmful to the operation of copper smelting. Anode slimes are produced from the electro-refining of the impure anodes, which contain most of the noble metals present in the ores, as well as the secondary resources. Therefore, it is important to develop an efficient process to recover noble metals with high purity from the anode slimes.

Anode slimes are generally treated by systems such as cyanide [6], halide [7], and biotechnologies [8]. The noble metal ions in the leaching solution are separated by precipitation [9], activated carbon adsorption [10,11], ion exchange [12,13], and solvent extraction [14–17]. Among these methods, solvent extraction has been employed in commercial plants for a long time. In the case of solvent extraction, some kinds of ionic liquids have been investigated for noble metal separation in the laboratory scale [15,18–20]. Considering the cost of ionic liquids and the possibility of their application

in the industrial scale, solvent extraction with commercial extractants has more advantages, in terms of commercial scale applications. Among the noble metals, the solubility of silver metal in a HCl solution is very low and other leaching solutions like $HNO_3$ and $H_2SO_4$ are firstly employed to dissolve silver metal [21,22]. In general, ruthenium(Ru) and osmium(Os) are separated from other platinum group metals (PGMs) by oxidative distillation [23,24].

Plenty of work has been reported on the separation of two or three noble metal ions by solvent extraction from a hydrochloric acid solution [25–29]. After the recovery of ruthenium and osmium by oxidative distillation, six noble metals remain in the anode slimes. When silver is separated as silver chloride, by leaching with an HCl solution in the presence of some oxidizing agents, the remaining five noble metal ions (Au(III), Pt(IV), Pd(II), Ir(IV), and Rh(III)) are present in the leaching solution. Although several solvent extraction processes have been reported to separate these five noble metal ions [5,14,18,30], they have some disadvantages, such as incomplete separation among the metal ions in each solvent extraction step and the low extraction percentage of Au(III) and Pd(II) from concentrated acid solutions.

We have reported solvent extraction processes to separate Pt(IV)-Pd(II)-Ir(IV)-Rh(III) and Au(III)-Pt(IV)-Pd(III) from hydrochloric acid solutions [26–29]. According to these works, Pd(II) can be selectively extracted over Pt(IV), by LIX 63, from a concentrated hydrochloric acid solution. Extraction of Au(III)-Pd(II)-Pt(IV), with Cyanex 272, from a concentrated hydrochloric acid solution results in the selective extraction of Au(III). Moreover, Au(III) has been successfully separated from Ag(I) and other base metal ions, by Cyanex 272, from the leaching solution of anode slimes [31]. Therefore, it is reasonable that a combination of the above two solvent extraction systems (Cyanex 272 and LIX 63) would lead to selective extraction of Au(III) and Pd(II) from the other noble metal ions. In order to verify this proposed process, the data on the mass balance and the purity of the obtained solution are necessary. For this purpose, solvent extraction experiments were done in a synthetic HCl solution containing Au(III), Pd(II), Pt(IV), Ir(IV) and Rh(III). In this work, gold was successfully separated over the other four metals by solvent extraction, with Cyanex 272, from the hydrochloric acid concentration, from 0.5 M to 9 M. Separation of the other four metals, by LIX 63, TBP, and Aliquat 336, was tried on the basis of the obtained conditions reported in the previous work [28]. Compared with the previous processes for separating the five metals, the process reported in this work does not need a scrubbing stage. The mass balance, purity, and recovery of each metal from the synthetic solution were obtained from the process proposed in this work.

## 2. Experimental

### 2.1. Chemicals and Reagents

The synthetic solution was prepared by dissolving certain amounts of $HAuCl_4$ (30 wt.% in dilute HCl, Sigma-Aldrich, St. Louis, MO, USA), $H_2IrCl_6$ (99.5%, Alfa-Aesar, Ward Hill, MA, USA), $RhCl_3$ (99.9%, Alfa-Aesar, Ward Hill, MA, USA), $PtCl_4$ (99.9%, Alfa-Aesar, Ward Hill, MA, USA), and $PdCl_2$ (99.9%, Alfa-Aesar, Ward Hill, MA, USA) in a HCl solution with the required concentration. Extractants employed in this work were 2-ethylhexyl phosphonic acid mono-2-ethylhexyl ester (Cyanex 272, 85%, Solvay Cytec Industries, Woodland Park, NJ, USA), 5,8-diethyl-7-hydroxydodecan-6-one oxime (LIX 63, 70%, BASF Co., Ludwigshafen, Germany), tributyl phosphate (TBP, 98%, Yakuri Pure Chemical Co., Ltd., Kyoto, Japan), and tricaprylmethylammonium chloride (Aliquat 336, 100%, BASF Co., Ludwigshafen, Germany). All the chemicals were of analytical grade. Commercial grade kerosene (Dea Jung Chemical Co., Siheung, Korea) was used as a diluent. In the whole experiment, the concentration of the metals in the synthetic solution was fixed at 100 mg/L.

### 2.2. Experimental Process

Solvent extraction experiments were performed in a 50 mL screw-cap bottle. Equal volume of organic and aqueous phase (10 mL) was shaken for 30 min by employing a wrist action shaker. After

shaking, the two phases were separated in a separate funnel. All the experiments were done at room temperature. The concentration of the metals in aqueous phase was measured by ICP-OES (Inductively coupled plasma-optical emission spectrometer, Spectro Arcos) and the metal concentration in the organic phase was calculated by mass balance. The extraction and stripping percentage of a metal was calculated by the following Equations (1) and (2). The error in the extraction and stripping percentage reported in this work was within ±5%.

$$\text{Extraction percentage} = \frac{\text{Equilbrium mass of metal in the organic}}{\text{Initial mass of metal in the aqueous}} \times 100\%, \tag{1}$$

$$\text{Stripping percentage} = \frac{\text{Equilibrium mass of metal in the aqueous}}{\text{Initial mass of metal in the organic}} \times 100\%. \tag{2}$$

## 3. Results and Discussion

### 3.1. Effect of HCl Concentration on the Extraction Au(III) by Cyanex 272

In general, the acidity of a solution has a great effect on the extraction behavior of the metal ions. Therefore, it is necessary to investigate the effect of HCl concentration on the selective extraction of Au(III), by Cyanex 272, in the presence of other noble metal ions. For this purpose, HCl concentration of the synthetic solutions containing Au(III), Pd(II), Pt(IV), Rh(III), and Ir(IV) varied from 0.5 M to 9 M. During the experiments, the concentration of Cyanex 272 was fixed at 0.2 M. The results in Figure 1 show that the extraction of Au(III) increased slowly, from 80% to 99.9%, as the HCl concentration increased and remained constant from 5 M to 9 M of HCl concentration. All the other metals, except Au(III), remained in the aqueous solution at all conditions employed in this work. Our results indicate that it is possible to extract only Au(III) over Pd(II), Pt(IV), Ir(IV), and Rh(III) in the HCl concentration range from 0.5 M to 9 M. Since no other metal ions except Au(III) were extracted, scrubbing is not necessary, which would be of immense value in commercial applications.

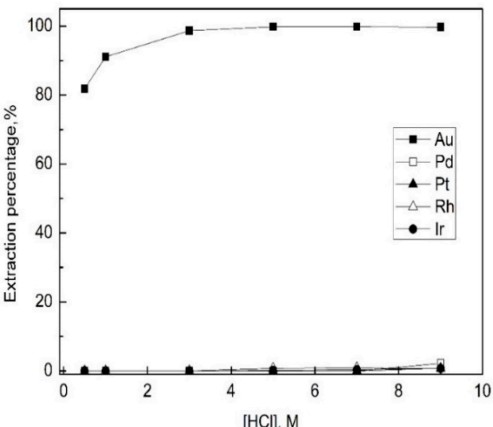

**Figure 1.** Effect of HCl concentration on the extraction of metals in a HCl solution containing Au(III), Pd(II), Pt(IV), Rh(III), and Ir(IV). ([HCl] = 0.5–9 M, [Cyanex 272] = 0.2 M, O/A = 1, [Au(III)] = [Pd(II)] = [Pt(IV)] = [Rh(III)] = [Ir(IV)] = 100 mg/L).

### 3.2. Effect of Cyanex 272 Concentration on the Extraction of Au(III)

Figure 1 shows that Au(III) was completely extracted by 0.2 M Cyanex 272 when HCl concentration was higher than 3 M. The effect of Cyanex 272 concentration on Au(III) extraction was investigated in a 5 M HCl solution (see Figure 2). In the course of extraction experiments, Cyanex 272 concentration varied from 0.01 to 0.25 M. When Cyanex 272 concentration increased from 0.01 to 0.05 M, the extraction of Au(III) rapidly increased from 42% to 93% and then slowly rose to 99.9% and remained at a constant value with the further increase of Cyanex 272 concentration. In these experiments, only Au(III) was

extracted and, thus, it was possible to separate Au(III) from the 4 PGMs present in the solution. Therefore, 0.2 M Cyanex 272 was selected as the optimum concentration for the separation of 100 mg/L Au(III) from other metal ions.

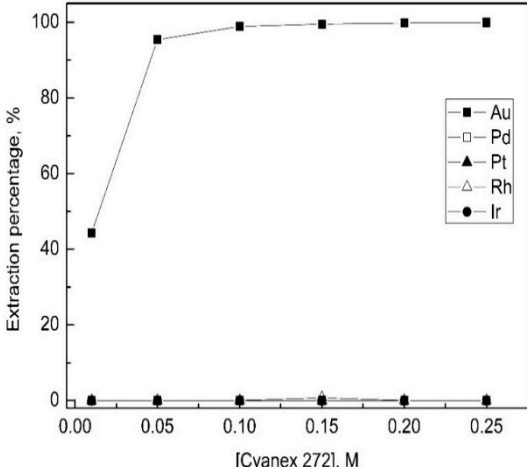

**Figure 2.** Effect of Cyanex 272 concentration on the extraction of metals containing Au(III), Pd(II), Pt(IV), Rh(III), and Ir(IV). ([HCl] = 5 M, [Au(III)] = [Pd(II)] = [Pt(IV)] = [Rh(III)] = [Ir(IV)] = 100 mg/L).

The composition of the anode slimes resulting from smelting depends on the nature of the ores and the secondary resources. Our data clearly indicates that only Au(III) is to be extracted by Cyanex 272 when the concentration of HCl in the leaching solution containing Pt(IV), Pd(II), Ir(IV), and Rh(III) is higher than 0.5 M. Therefore, pure Au(III) can be obtained by selective extraction, resulting in a much-improved efficiency, compared to the reported solvent extraction processes.

In our previous work [29,31], Au(III) concentration was varied from 100 to 1000 mg/L in the presence of Pd(II) and Pt(IV). Even though the concentration ratio of Au(III) to Pd(II) was varied from unity to 10, only Au(III) was extracted over Pt(IV) and Pd(II), by Cyanex 272. Moreover, the presence of some base metals has little effect on the extraction of Au(III), by Cyanex 272, from the leaching solution. Although 0.2 M Cyanex 272 was selected as an optimum concentration for the selective extraction of 100 mg/L Au(III), the concentration of Cyanex 272 can be adjusted according to the concentration of Au(III) in the leaching solution because only Au(III) is extracted by Cyanex 272.

In a concentrated HCl solution (>4 M), the predominant species of Au(III), Pd(II), Pt(IV) and Ir(IV) are supposed to be $AuCl_4^{2-}$, $PdCl_4^{2-}$, $PtCl_6^{2-}$, and $IrCl_6^{2-}$, respectively [32–34]. The distribution of Rh(III) is complicated and depends on the concentration of chloride ions. In a concentrated HCl solution, Rh(III) exists as either $RhCl_6^{3-}$ or $RhCl_5(H_2O)^{2-}$ [35]. Figures 1 and 2 imply that the difference on the charge densities of the above complexes could not explain the selective extraction of Au(III) by Cyanex 272 over the 4 PGMs. More fundamental work is needed for the identification of the extraction mechanism. The probable reaction has been proposed in a previous work as Equation (3) [29], as follows:

$$H^+ + AuCl_4^- + 2HA_{org} = [HAuCl_4 \cdot 2HA]_{org.} \tag{3}$$

### 3.3. Stripping of Au(III) from Loaded Cyanex 272

$NH_4Cl$, $(NH_4)_2S_2O_3$ and $(NH_2)_2CS$ are found to be effective in the stripping of Au(III) from the loaded Cyanex 272 [29,34]. On the basis of the stability and the cost of these reagents [36], stripping of Au(III) with $NH_4Cl$ and $(NH_2)_2CS$ was tried. For this purpose, the loaded 0.2 M Cyanex 272 was prepared by contacting the synthetic solution with 5 M HCl. More than 99.9% of Au(III) was extracted and, thus, the concentration of Au(III) in the loaded Cyanex 272 was around 99.9 mg/L. The concentration of $NH_4Cl$ and $(NH_2)_2CS$ was varied from 0.1 to 1.2 M and 0.1 to 0.5 M, respectively. The concentration of the two stripping agents did not affect the stripping percentage of Au(III) (see

Figure 3). More than 98.5% of Au(III) was stripped by (NH$_2$)$_2$CS, while about 70% of Au(III) was stripped by NH$_4$Cl, in the experimental range. A previous work reported that Au(III) is completely stripped by NH$_4$Cl from the loaded Cyanex 272, which contains Sn(II) [31]. Sn(II) can take part in a redox reaction between Sn(II) and Sn(IV), which facilitates the stripping of Au(III) from the loaded Cyanex 272.

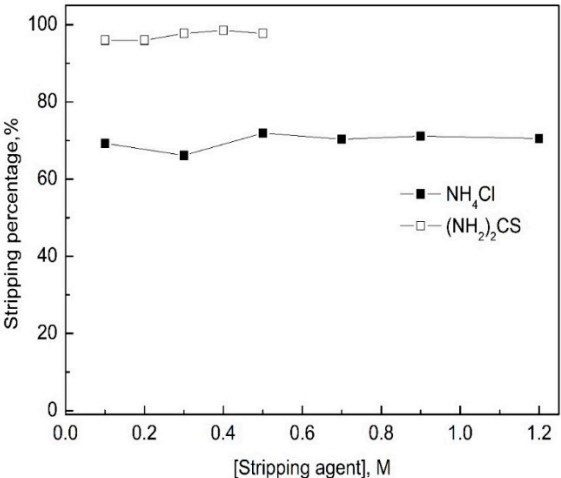

**Figure 3.** Stripping of Au(III) from the loaded Cyanex 272 by different stripping agents. ((Cyanex 272) = 0.2 M, Loaded (Au(III)) = 99.9 mg/L, (NH$_4$Cl) = 0.1–1.2 M, ((NH$_2$)$_2$CS) = 0.1–0.5 M).

*3.4. Integrated Procedure*

After selective extraction of Au(III) from the other 4 PGMs, Pd(II) can be selectively extracted from the raffinate with LIX 63, as reported in the separation of Pt(IV) and Pd(II) [29]. Then, Pt(IV), Ir(IV), and Rh(III) can be sequentially separated by the extraction steps reported in the literature [28].

Figure 4 shows the proposed process for the separation of noble metals from a 5 M HCl solution containing Au(III), Pd(II), Pt(IV), Rh(III), and Ir(IV). The mass balance of the metal ions during each step is listed in Table 1. First, Au(III) was separated by Cyanex 272 and the loaded Au(III) was completely stripped by (NH$_2$)$_2$CS. Based on the previous report [28], Pd(II), Pt(IV), Ir(IV), and Rh(III) were sequentially extracted by LIX 63, TBP, and Aliquat 336 from the Au(III) free raffinate, respectively. Meanwhile, the stripping of these noble metals was successfully accomplished. Therefore, the five noble metals in the synthetic solution can be separated in each step. According to Table 1, the recovery percentage of Au(III), Pd(II), Ir(IV), and Rh(III) was higher than 98% and that of Pt(IV) was 94%. Considering the recovery percentage and the purity of the respective solution of each noble metal, the process proposed in this work can be very efficient in the treatment of HCl leaching solutions of secondary resources containing noble metals. In general, the anode slimes resulting from electro-refining do not contain base metals. Since only Au(III) was extracted by Cyanex 272 over Pt(IV), Pd(II), Ir(IV), and Rh(III), this process can be applied to HCl leaching solutions with variable compositions of Au(III) and the PGMs.

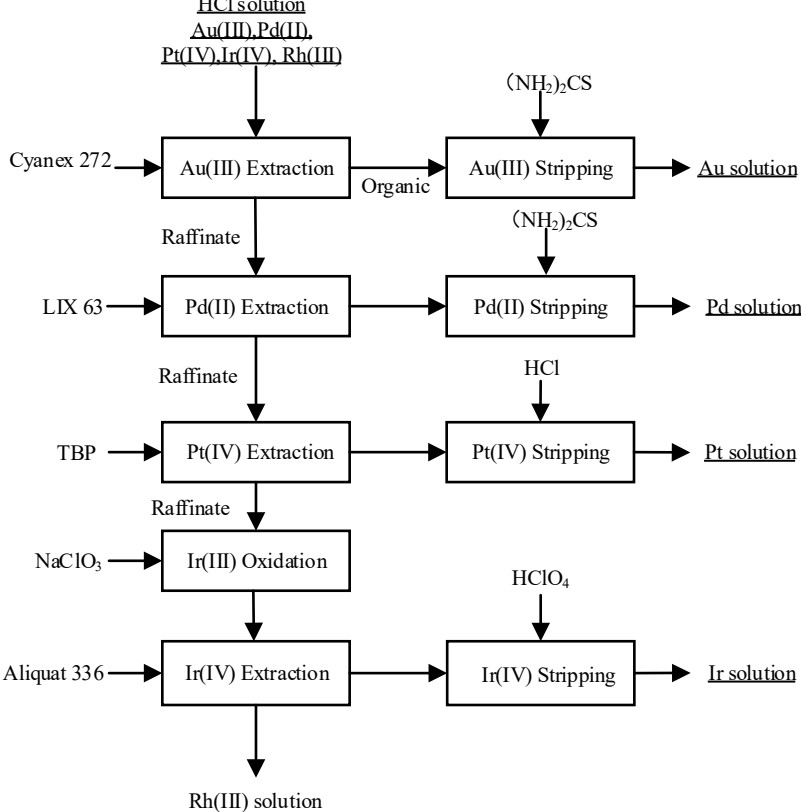

**Figure 4.** Integrated process for the separation of noble metals from a HCl solution by solvent extraction.

**Table 1.** Mass balance of metals in each step of the process proposed in this work.

| Process | Detail | Au(III) | Pd(II) | Pt(IV) | Ir(IV) | Rh(III) |
|---|---|---|---|---|---|---|
| **Feed solution** | Synthetic solution, 5 M HCl, mg/L | 100 | 100 | 100 | 100 | 100 |
| **Au(III) separation** | Extraction: 0.2 M Cyanex 272, O/A = 1, % | 99.9 | / | / | / | / |
| | Stripping: 0.3 M (NH$_2$)$_2$CS, A/O = 1, % | 99.9 | / | / | / | / |
| | Raffinate, mg/L | 0.01 | 100 | 100 | 100 | 100 |
| **Pd(III) separation** | Extraction: 0.05 M LIX63, O/A = 1, % | / | 95.7 | 0 | 0 | 0 |
| | Stripping: 0.1 M (NH$_2$)$_2$CS, A/O = 1, % | / | 99.9 | / | / | / |
| | Raffinate, mg/L | / | 6.05 | 100 | 100 | 100 |
| **Pt(IV) separation** | Extraction: 2 M TBP, O/A = 1, 2 stages % | 0 | 0 | 98.6 | 0 | 1.95 |
| | Stripping: 0.01 M HCl, O/A = 1, 3 stages, % | / | / | 95.7 | / | / |
| | Raffinate, mg/L | / | / | 1.3 | 100 | 100 |
| **Ir(IV) separation** | Extraction: 0.03 M Aliquat 336, 5% TBP, O/A = 1, % | / | / | / | 98.1 | / |
| | Stripping: 0.3 M HClO$_4$, A/O = 1, % | / | / | / | 99.9 | / |
| | Raffinate, mg/L | / | / | / | 5.7 | 100 |
| **Results** | Extraction percentage, % | 99.9 | 99.8 [1] | 98.6 [1] | 98.1 [1] | 99.9 |
| | Stripping percentage, % | 99.9 | 99.9 | 95.7 [1] | 99.9 | 99.9 |
| | Recovery, % | 99.8 | 99.7 | 94.4 | 98.1 | 99.9 |

[1] Pd(II) extraction: 2 stages; Pt(IV) extraction: 2 stages, stripping 3 stages; Ir(IV) extraction: 2 stages.

　　　Further work needs to be done to elucidate the extraction mechanism of Au(III) by Cyanex 272. Moreover, long term operation of the solvent extraction by employing real leaching solutions of anode slimes is necessary to verify this process.

## 4. Conclusions

A solvent extraction process was developed for the separation of noble metals from concentrated hydrochloric acid solutions containing Au(III), Pd(II), Pt(IV), Rh(III), and Ir(IV). In the process proposed in this work, Au(III), Pd(II), Pt(IV), Ir(IV), and Rh(III) were sequentially separated by solvent extraction. The optimum conditions for the extraction and stripping of the metal ions from the respective extractants were reported for a synthetic solution with 5 M HCl, where the concentration of all the metal ions was fixed at 100 mg/L. Successive employment of Cyanex 272, LIX 63, TBP, and Aliquat 336 resulted in a raffinate containing only Rh(III). The mass balance for the whole process indicated that the recovery percentage of the noble metal ions, except Pt(IV), was higher than 98%. This process could be applied to the recovery of noble metals with high purity from the hydrochloric acid leaching solutions of the secondary resources.

**Author Contributions:** M.S.L. designed the research and helped to analyze the data. W.D.X. performed the experiments and wrote the paper.

**Funding:** This work was supported by the Global Excellent Technology Innovation of the Korea Institute of Energy Technology Evaluation and Planning (KETEP), granted financial resource from the Ministry of Trade, Industry & Energy, Republic of Korea (No. 20165010100880).

**Acknowledgments:** We express sincere thanks to the Korea Basic Science Institute (KBSI), Gwangju branch for providing ICP-OES data.

**Conflicts of Interest:** The authors declare no conflict of interest.

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
