# Peer review of "A Process for the Separation of Noble Metals from HCl Liquor Containing Gold(III), Palladium(II), Platinum(IV), Rhodium(III), and Iridium(IV) by Solvent Extraction"

_processes, doi:10.3390/pr7050243_

Reviewer 1 Report

In my opinion, this paper presents relatively useful, interesting, and industrially relevant results and it can be accepted for publication as long as the authors read it several times and correct many typos/grammatical errors present in the paper.

Author Response

Thank you for your comments. Since we are not native English speakers, there should be some errors and awkward expressions in English. We have tried to improve the quality of English and correct some errors. The English of the manuscript has been revised.

Reviewer 2 Report

The manuscript provides a crucial approach to separate noble metals from anode slimes in order to recycle and reuse them using solvent extraction. The background of the study has been explained very well, the experimental section has been presented thoroughly. However, some improvements are necessary in order to provide a total picture and make it clear and therefore, the manuscript should be reconsidered for publication after major revision.

1) Please use full forms of the chemicals or terms the first time they are used in the manuscript. e.g. chemicals mentioned in section 2.1, TBP, etc.

2) In addition to leaching solution and extractant concentration, there are a number of process parameters (e.g. feed solution pH) that play a critical role in the solvent extration process and the rate of extraction. A discussion and some studies that would provide a correlation of these parameters with extraction rate, recovery, and purity of noble metals, should be provided. 

3) Section 3.3, Line 150: "the presence of Sn(II) in Cyanex 272 have some influence on the stripping of Au(III)." It would be good to provide more discussion on this influence (possible reasons, mechanisms).

4) There are some anomalies observed in the mass balance table. After Pt(IV) separation, the stripping solution has 65% Pt. If only Pt is back extracted, shouldn't it be 99.9/100% in the stripping solution?

5) It would be useful to show the trend in extraction with a higher concentration of noble metals instead of using only 100 ppm. Understanding the effect of concentration of other metals on the extraction of Au would be important.

Author Response

1) Please use full forms of the chemicals or terms the first time they are used in the manuscript. e.g. chemicals mentioned in section 2.1, TBP, etc.

Ans: The full name of the chemicals has been added to the manuscript.

Modification: line 80-83

Extractants employed in this work were 2-ethylhexyl phosphonic acid mono-2-ethylhexyl ester (Cyanex 272, 85%, Solvay Cytec Industries, Canada), 5,8-diethyl-7-hydroxydodecan-6-one oxime (LIX 63, 70%, BASF Co., USA), tributyl phosphate (TBP, 98%, Yakuri Pure Chemical Co., Ltd. Korea), tricaprylmethylammonium chloride (Aliquat 336, 100%, BASF Co., USA).

2) In addition to leaching solution and extractant concentration, there are a number of process parameters (e.g. feed solution pH) that play a critical role in the solvent extraction process and the rate of extraction. A discussion and some studies that would provide a correlation of these parameters with extraction rate, recovery, and purity of noble metals, should be provided.

Ans: We agree with the reviewer’s comments. Solution pH has an important effect on the solvent extraction of metals.

In general, around 5 M HCl solution in presence of an oxidizing agent is employed as a leaching agent for the noble metals. Therefore, effect of HCl concentration on the selective extraction of Au(III) from other noble metal ions was first investigated in this work. According to Fig. 1., only Au(III) was extracted by Cyanex 272 in the HCl concentration range from 0.5 M to 9 M. Since other noble metal ions (Pt(IV), Pd(II), Ir(IV) and Rh(III)) remained in the raffinate, the purity of Au(III) in the loaded Cyanex 272 was higher than 99.9%, which is reflected in Table 1 of the manuscript.

According to our previous work, HCl concentration does not affect the solvent extraction rate and equilibrium can be obtained within 30 mins.

The manuscript has been revised as follows.

Modification: line 100-104

In general, the acidity of a solution has a great effect on the extraction behavior of the metal ions. Therefore, it is necessary to investigate the effect of HCl concentration on the selective extraction of Au(III) by Cyanex 272 in the presence of other noble metal ions. For this purpose, HCl concentration of the synthetic solutions containing Au(III), Pd(II), Pt(IV), Rh(III) and Ir(IV) was varied from 0.5 M to 9 M. During the experiments,

3) Section 3.3, Line 150: "the presence of Sn(II) in Cyanex 272 have some influence on the stripping of Au(III)." It would be good to provide more discussion on this influence (possible reasons, mechanisms).

Ans: The effect of Sn(II) on the stripping of Au(III) from the loaded Cyanex 272 has been revised as follows.

Modification: line 157-160

NH4Cl in the experimental range. It has been reported that Au(III) is completely stripped by NH4Cl from the loaded Cyanex 272 which contains Sn(II) in a previous work [31]. Sn(II) can take part in a redox reaction between Sn(II) and Sn(IV), which facilitates the stripping of Au(III) from the loaded Cyanex 272.

4) There are some anomalies observed in the mass balance table. After Pt(IV) separation, the stripping solution has 65% Pt. If only Pt is back extracted, shouldn't it be 99.9/100% in the stripping solution?

Ans: Thanks for your comments.  

The stripping percentage of Pt(IV) by 0.01 M HCl was 65% at an O/A ratio of unity. McCabe-Thiele stripping diagram indicated that 3 stages were necessary to strip most of the Pt(IV) from the loaded TBP. Therefore, batch simulation experiments for the 3 stage counter-current stripping were done. After 3 stage, the stripping percentage of Pt(IV) was 95.7%.

We corrected Table 1 to show this.

5) It would be useful to show the trend in extraction with a higher concentration of noble metals instead of using only 100 ppm. Understanding the effect of concentration of other metals on the extraction of Au would be important.

Ans: Thanks for the comments.

Generally, the concentration of metal ions affects the extraction behavior of metal ions. In our previous work, we investigated the effect of Au(III) concentration on the extraction behavior from the solution containing Pd(II) and Pt(IV). According to the obtained results, only Au(III) was extracted by Cyanex 272 when the concentration ratio of Au(III) to Pt(IV) or Pd(II) was within 10. Moreover, the effect of the presence of some base metals was investigated. At these conditions, only Au(III) was extracted.

The manuscript has been revised as follows.

Modified parts: line 130-136

In our previous work[29,31], Au(III) concentration was varied from 100 to 1000 mg/L in the presence of Pd(II) and Pt(IV). Even though the concentration ratio of Au(III) to Pd(II) was varied from unity to 10, only Au(III) was extracted over Pt(IV) and Pd(II) by Cyanex 272. Moreover, the presence of some base metals has little effect on the extraction of Au(III) by Cyanex 272 from the leaching solution. Although 0.2 M Cyanex 272 was selected as an optimum concentration for the selective extraction of 100 mg/L Au(III), the concentration of Cyanex 272 can be adjusted according to the concentration of Au(III) in the leaching solution because only Au(III) is extracted by Cyanex 272. 

line 179-182

In general, the anode slimes resulted from electro-refining do not contain base metals. Since only Au(III) was extracted by Cyanex 272 over Pt(IV), Pd(II), Ir(IV) and Rh(III), this process can be applied to HCl leaching solutions with variable composition of Au(III) and the PGMs.

Round  2

Reviewer 2 Report

The authors have modified the manuscript as per the suggestions and I think it provides a clear picture for solvent extraction process of noble metals from anode slimes.